

# Controlled temperature contrasts of three native and one highly invasive annual plant species in California

Mario Zuliani, Stephanie Haas-Desmarais, Laura Brussa, Jessica Cunsolo, Angela Zuliani and Christopher J. Lortie

Biology, York University, Toronto, Ontario, Canada

## ABSTRACT

Plant responses to changes in temperature can be a key factor in predicting the presence and managing invasive plant species while conserving resident native plant species in dryland ecosystems. Climate can influence germination, establishment, and seedling biomass of both native and invasive plant species. We tested the hypothesis that common and widely distributed native and an invasive plant species in dryland ecosystems in California respond differently to increasing temperatures. To test this, we examined the effects of temperature variation on germination, establishment, and *per capita* seedling biomass of three native and one invasive plant species (*Bromus rubens*) in independent 6 week growth trial experiments in a controlled greenhouse. Higher relative temperatures reduced the germination and establishment of the tested invasive species and two tested native species, however, *per capita* biomass was not significantly affected. Specifically, germination and establishment of the invasive species *B. rubens* and the native species *Phacelia tanacetifolia* was significantly reduced. This invasive species can often outcompete natives, but increasing temperature could potentially shift the balance between the germination and establishment of natives. A warming climate will likely have negative impacts on native annual plant species in California tested here because increasing temperatures can co-occur with drought. This study shows that our tested native annual plant species tested here have some resilience to relatively significant increases in temperature, and this can favor at least one native species relative to at least one highly noxious invasive plant species.

## INTRODUCTION

Dryland ecosystems including shrublands, deserts, and grasslands are crucial to plant communities globally as they support endemic species, thus influencing global biodiversity (*Sharafatmandrad & Khosravi Mashizi, 2021*; *Lucero et al., 2022*). With increasing abiotic stressors, including rising temperatures and prolonged periods of minimal precipitation, these ecosystems are experiencing more frequent and severe degradation (*King & Hobbs, 2006*; *Huang et al., 2020*). It is imperative to determine the impacts these conditions will have on dryland plant community composition and diversity. Increasing temperature (*Shah et al., 2011*), drought events (*Niu, Rodriguez & Wang, 2006*; *Verwijmeren et al.,*

Corresponding author
Mario Zuliani,
zulianimario96@gmail.com

*2019*), and other abiotic disturbances (*Potts et al., 2012*), have overall negative impacts on many plant species, but can particularly create conditions more favorable for invasives over natives in dryland ecosystems (*Niu, Rodriguez & Wang, 2006*; *Cowles et al., 2018*; *Moore, Stow & Kearney, 2018*). Ecosystems, such as the Carrizo Plain National Monument (35.11982, −119.62853) in Southern California are highly invaded by plant species, such as *Bromus rubens* and *Schismus barbatus* (*Lucero et al., 2019*), and subject to extreme weather conditions, including megadroughts (*Westphal et al., 2016*; *Lortie et al., 2021*). The mean annual precipitation within this region can reach 250 mm, however this area can experience extensive periods of extreme drought (*Prugh et al., 2018*). The variation in temperatures across dryland ecosystems can have strong negative influences on plant establishment, germination, and productivity in the form of biomass, fruiting, and flowering (*Rivas-Arancibia et al., 2006*; *Ebrahimi & Eslami, 2012*; *Hatfield & Prueger, 2015*). For instance, plant species can experience thermoinhibition, defined as the inability for seeds to germinate at relatively high temperatures (*Vleeshouwers, Bouwmeester & Karssen, 1995*; *Hills & van Staden, 2003*; *Toh et al., 2012*), which can increase seed dormancy, preventing the initial stages of germination. This thermoinhibition gives species with larger temperature niches a significant advantage in terms of germination and establishment by delaying germination of potential competitors (*Toh et al., 2012*; *da Silva et al., 2017*). Some invasive plant species have developed mechanisms to bypass thermoinhibition, giving them a potential competitive advantage over native plant species because they germinate sooner (*Baskin, 1998*; *Urbanova & Leubner-Metzger, 2018*; *Bhatt et al., 2023*). This can result in native species being outcompeted for space and resources at some temperatures (*Dukes & Mooney, 1999*; *Gioria, Pyšek & Moravcova, 2012*). While other factors can influence dryland plant species, temperature can be a key factor in determining the outcome of native-invasive competition (*Verlinden, De Boeck & Nijs, 2014*; *Lu et al., 2016*). Understanding how these constantly changing dryland ecosystems influence plant species composition provides key insights into the establishment of different plant species. Temperature is but one key factor, but is highly variable in these dryland systems, potentially shaping community composition (*Bertrand et al., 2011*).

The relative growth rate of plant species differs and responds uniquely to key environmental factors that signal when to grow and at what rate (*Bareke, 2018*). There are several early life-stage performance measures associated with plant species including: germination—the appearance of an embryo through the surrounding structure (*Nonogaki, Bassel & Bewley, 2010*), establishment—seedlings are established typically at 2–3 weeks for most semi-arid annuals (*Aronson et al., 1992*; *Pik et al., 2020*), and seedling biomass—a proxy for future individual performance (*Proulx et al., 2015*; *Liczner et al., 2019*; *Pik et al., 2020*). These early-life stage performance measures can be used to predict species distribution and abundance because they set the framework for plant-plant interactions when individuals are established (*Pyke & Archer, 1991*; *Theoharides & Dukes, 2007*). Knowledge on seed and seedling dynamics in particular will help managers optimize native planting strategies and increase the understanding of how these species interact with invasive species in the ecosystem (*Costantini et al., 2016*; *James & Carrick, 2016*). Temperature can influence early-life stage measures, likely increasing all factors initially,
provided there is sufficient soil moisture and nutrients. However, increasing temperatures can become a stressor depending on a species' climate niche (*Vázquez-Ramírez & Venn, 2021*). To ameliorate these high abiotic stressors, such as extreme temperatures, areas with more favorable microclimatic conditions, such as shrub canopies and artificial structures that provide shade, can positively influence plant association and establishment (*Filazzola et al., 2019*; *Roque Marca, López & Naoki, 2021*; *Zuliani et al., 2024a*). Foundational shrubs within these dryland ecosystems can buffer high temperatures, promoting plant species associations in dryland ecosystems (*Lortie et al., 2024*). This suggests that plant species within drylands favor conditions not directly exposed to high temperatures, such as the microclimates underneath shrub canopies (*Pugnaire, Armas & Valladares, 2004*). These opportunities for stress amelioration, through the provision of relatively more habitable microclimates and increasing soil nutrient availability, are utilized by both native and invasive plant species and can provide more favorable early-life stage growing conditions for native species (*Muñoz-Rojas et al., 2016*; *Lucero et al., 2019*). Managers can leverage these more favorable microclimatic conditions by focusing planting efforts in areas that are potentially more favorable for native species, promoting their earlier establishment and potentially allowing them to outcompete invasive plants (*Bossard, Randall & Hoshovsky, 2000*; *Liczner et al., 2019*).

Invasive plant species can influence the structure and composition of plant communities in ecosystems globally (*Laughlin & Abella, 2007*; *Flory & Clay, 2009*; *Pik et al., 2020*; *Szumańska et al., 2021*). These invasive plant species disrupt local communities through the decline and degradation of biodiversity, while negatively impacting ecosystem functions including soil fertility and water availability (*Grice, 2006*; *Maestre et al., 2016*; *Milanović et al., 2020*). The negative impacts of invasive plants can also extend to animal communities by displacing habitats (*Beck et al., 2008*), reducing foraging behavior of both livestock and wildlife (*Brunson & Tanaka, 2011*), and impeding the movement of some animal species (*Stewart et al., 2021*). Some invasive species are resilient to high temperatures including many species in the genus *Bromus* (*Abella et al., 2011*; *Clements & Ditommaso, 2011*). These brome species pose a significant threat to many dryland ecosystems because they promote increased fire frequency and intensity (*Monty, Brown & Johnston, 2013*; *Fenesi et al., 2016*), displace native vegetation reducing their overall biodiversity and biomass (*Gill et al., 2018*; *Palit & DeKeyser, 2022*), and negatively influence wildlife through habitat loss (*Freeman et al., 2014*; *Germino et al., 2016*). In dryland ecosystems, invasive plants can be resilient to drought-like conditions, potentially out-performing native species (*Ali & Bucher, 2022*). This extended climate envelope can help invasive species outcompete natives, as they have a higher probability of establishing successful populations (*Bradley, Wilcove & Oppenheimer, 2010*; *Hou et al., 2014*). Central and Southern California drylands are both highly invaded by many exotic species, including brome, and are resident to a high diversity of native annual plant species (*Seabloom et al., 2003*; *Fisher, Del Pinto & Fisher, 2020*). Thus, the importance of variation in temperature on these plant species and common communities is key to conservation (*Lucero et al., 2022*). Temperature changes in semi-arid ecosystems, alongside increasing drought frequency (*Cherwin & Knapp, 2012*), suggests that we need to better understand

species-specificity in response to changes in temperature (*Parmesan & Hanley, 2015*). This is important at local and regional sites in predicting if species will spread or invade into other regions when temperatures increase (*Wallingford et al., 2020*; *Ali & Bucher, 2022*). Further understanding how these invasive species are impacted by their climate will provide valuable insight both for the management of these species and for predicting potential scenarios for community assembly of native *vs* invasive with a changing climate.

The purpose of this study is to determine if increasing temperatures influence common natives and an invasive plant species local to the arid/semi-arid ecosystems of Southern California. To test this, we used a controlled greenhouse to conduct temperature trials on three native and one highly invasive plant species independently. We examined the hypothesis that temperature directly influences key early-life stage performance measures in these semi-arid annual plant species, with their responses being highly species-specific. We tested the following predictions:

1) Increasing long-term temperatures associated with GBIF observation for each species negatively influences the species observations.
2) The invasive species *Bromus rubens* is most likely to benefit from increasing experimental temperatures because this species has been reported to germinate and establish at extreme temperatures (*Bykova & Sage, 2012*; *Bykova, 2014*).
3) Early-life stage measures for all native species will be negatively impacted by increasing experimental temperatures.

## Methods

A series of independent controlled temperature experiments were conducted to test the impacts of increasing temperatures on early-life stage performances of Southern California plant species. A total of three native and one highly invasive plant species were tested independently for 6 weeks each to determine if increasing temperature would negatively influence the germination, establishment, and mean *per capita* seedling biomass. Global Biodiversity Information Facility (GBIF) data were also compiled for each species located within the central drylands of Southern California to determine the climate envelope for these species from reported occurrences (Fig. S1; *Global Biodiversity Information Facility (GBIF), 2023*; *Zuliani et al., 2024b*).

## Study species

*Bromus madritensis rubens* (hereafter referred to as *Bromus rubens*) is an annual grass species native to regions of Southern Europe, Southeast Asia, and Northern Africa (*Rauber, Cipriotti & Collantes, 2014*). *Bromus rubens* is a highly invasive species in Southern California because it is rapidly invading large portions of the Mojave, Sonora, and Great Basin Deserts, as well as semi-arid grasslands, such as the Carrizo Plain National Monument (*Ogle, Reiners & Gerow, 2003*; *Abella et al., 2012*; *Curtis & Bradley, 2015*). This invasive species has a blooming season between February and June, and can quickly dominate local plant communities, outcompeting them for nutrients and light, while also

altering microhabitats (*Hamilton, Holzapfel & Mahall, 1999*; *Brooks, 2000*; *Gioria & Osborne, 2014*). *Bromus rubens* typically grow to heights of 16–40 cm tall with a red coloration to the upper most seed-dense head and have a seed viability of approximately 95% (*Wu & Jain, 1978*; *Jurand & Abella, 2013*). This species germination rate is more rapid than native annuals in dryland ecosystems, as a small amount of precipitation can awaken seeds from dormancy (*Salo, 2004*). *Bromus rubens* can survive at temperature extremes ranging from 10–36 °C (*Bykova & Sage, 2012*; *Bykova, 2014*).

*Layia platyglossa* (Fisch. & C.A. Mey.) (*Asteraceae*), *Phacelia tanacetifolia* Benth. (*Boraginaceae*), and *Salvia columbariae* Benth. (*Lamiaceae*), are three native species common in various arid and semi-arid regions within Southern California (*Buck-Diaz & Evens, 2011*). *Layia platyglossa*, also known as tidy tips, ranges in size from 45–60 cm, and has a blooming season ranging from March to June with temperatures ranging between 21–40 °C, with a potential seed viability of more than 80% (*Hobbs & Mooney, 1985*; *Christensen, 2000*; *Marty & BassiriRad, 2014*). *Phacelia tanacetifolia* has an average plant height of around 42 cm and can survive late spring and early summer temperatures, having an optimal growth temperature of around 30 °C, and a blooming season between April to June (*Yıldız, 2022*). *Phacelia* species typically display a seed viability of approximately 98% (*Cavieres & Arroyo, 2000*). The height of *S. columbariae* ranges from 10–50 cm, can survive temperatures ranging from 20–35 °C, have a blooming season between March and July, with related *Salvia* species having a seed viability of approximately 98% (*Adams, Wall & Garcia, 2005*; *Al-Turki & Baskin, 2017*; *Grimes et al., 2020*). These species commonly co-occur with *B. rubens* in North America (*Horn & St. Clair, 2017*). They are of particular interest as they have been disrupted by the invasion of *B. rubens* (*Liczner et al., 2019*; *Arroyo et al., 2021*). *Layia platyglossa*, *P. tanacetifolia*, and *S. columbariae*, are essential resources for both herbivorous species and native pollinators (*Ferrero et al., 2013*; *Braun & Lortie, 2019*; *Bishop et al., 2020*) and may compete with *B. rubens* (*Pik et al., 2020*).

*Bromus rubens* seeds were collected in the field within Southern California at the Wind Wolves Preserve (34.9929, −119.1832) within a 1.6 km radius. All native California seeds were purchased through Outsidepride, where seeds are produced in a greenhouse setting. All seeds were ordered as needed and received within 2 days of conducting each species' seed trials. Seeds were stored in ziplock bags and boxed to avoid direct sunlight exposure and reduce humidity exposure (*Suma et al., 2013*). Seeds were then kept at a constant temperature of around 8 °C. While all species tested can be found in areas outside of the drylands of Southern California, determining how increasing temperatures, in these specific ecosystems, influence early-life stages can provide insight on these species' ability to establish in more arid conditions.

## Experimental design

Temperature trials were done in a climate-controlled facility in Toronto, Canada. The effects of increasing temperatures on the germination, establishment (the germination and growth of an individual), and seedling biomass (the total weight of individual plants) of three California Native plant species (*S. columbariae, L. platyglossa, P. tanacetifolia*) and one invasive plant species (*B. rubens*) were tested for several key design reasons. Each

species was tested independently to maximize replication with our experiments consisting of 70 pots per treatment situated non-randomly (*Rogers et al., 2021*). Each species was also tested separately to ensure independence between species and prevent interspecific interactions (*Morris et al., 2007*; *Abdala-Roberts & Moreira, 2024*). Finally, even temperature in a controlled environment can vary, and thus while there were variations in temperature applied to each species, there was some overlap. Once a 6-week trial was completed, the next trial commenced. Here, germination is defined as the visible emergence of early-stage seed development of 0.1 mm (*Porceddu et al., 2013*). Species were tested in 10 cm diameter pots (1,400 cm$^3$ in soil volume). A total of 40 seeds for each species were sown independently, per pot, approximately 5 cm below the soil surface (*Lortie, Ghazian & Zuliani, 2022*). A total of 210 pots were utilized per trial with 70 pots designated across three treatments. To approximate the soil of California arid/semi-arid ecosystems, we mixed potting soil with coarse sand at a 1:1 ratio (*Pik et al., 2020*). Once seeds were sown, approximately 75 mL of water was measured for each pot once every 2 weeks. Large heat lamps (Simple Deluxe, Duarte, CA, USA) and Garpsen 315 LED Plant Lights (Kingbo, Guzhen, China) were positioned to fully cover the pots. Ambient temperature in the greenhouse averaged 21 °C. Heat lamps with lower wattage bulbs (40 W) produced an average temperature of 23 °C, medium wattage bulbs (60 W) produced an average of 26 °C, while higher wattage bulbs (100 W) averaged 31 °C. Heat lamps contained the same wattage bulbs per treatment to simulate areas of low, medium, and high temperature. All temperature and heat lamps were set on a 12 h timer to simulate light/dark cycles. An LI-250A light meter was used to measure the light intensity of all lightbulbs in µmol/m$^2$s. Local ambient temperature was recorded using a total of nine OMEGA pendant loggers suspended 10 cm on a stake in a pot chosen at random (https://www.omega.ca/en/data-acquisition/data-loggers/temperature-and-humidity-data-loggers/om-90-series/p/OM-92; Omega, Michigan City, IN, USA). Germination measurements were taken weekly to determine the number of seedlings that had emerged. Proportion of germinated seeds were calculated as the number of germinated seeds divided by the total number of sown seeds per pot (*Pik et al., 2020*). Establishment measurements were recorded as the total number of individual seedlings present per pot at the end of 6 weeks. Proportion established was calculated as the number of established seedlings divided by the total number of sown seeds per pot (*Pik et al., 2020*). All seedlings in each pot were dried in a Yamato Mechanical Convection Oven DKN900 for 72 h at 62 °C and subsequently weighed to measure total seedling dry biomass per pot. *Per capita* biomass was calculated as the total biomass divided by the total number of established individuals per pot (*Pik et al., 2020*). Any seeds that did not germinate during their 6 week trial were not included in this biomass estimate.

## Temperature validation

The temperatures used to conduct the greenhouse trials were selected based on the temperatures associated with the reported occurrences for each species within the region of Southern California. Global Biodiversity Information Facility (GBIF) was used to compile the occurrence data for all four species tested in this study (*Global Biodiversity Information*

*Facility (GBIF), 2023*). Climatic data were then gathered from WorldClim with a 0.5 min resolution (~1 sq-km). The climatic data taken from WorldClim was cropped to correspond to the GBIF data including the minimum and maximum observed latitudes and longitudes reported for each species (Latitude: 35.28531–36.56198, Longitude: −120.2184 to −116.7162). Climatic and occurrence data were then combined to both validate the temperature used in the greenhouse trials to estimate the current climatic niche of each species (*Pender et al., 2019*). Mean and max monthly temperatures were then derived from the WorldClim data to match the extent of the spatial occurrences of each species (*Pearson et al., 2002*). These estimates were used to infer the climate niche for each species (https://github.com/RS-eco/climateNiche; *Schweiger et al., 2014*). These data were used to generate distribution climate occurrences for each species independently.

## Statistical analysis

All statistics and models were done using the programming language R version 4.2.1 (*R Core Team, 2023*). WorldClim and GBIF data were fitted to a linear regression with a poisson distribution to estimate predicted values for climate envelopes of each plant species independently (*Alhajeri & Fourcade, 2019*). The mean temperature of the warmest quarter and the max temperature of the warmest month (*Fick & Hijmans, 2017*) associated with reported observations for each species were tested with linear regressions. Each species was experimentally tested independently, and the temperatures for each treatment across trials varied slightly. Thus, meta-analyses were used to first test for the heterogeneity between species in the global models for each response variable (*Hardy & Thompson, 1998*). The statistical significance of species and mean temperature as moderators was tested as a non-random effect using the rma function in the 'metafor' package (*Viechtbauer, 2010*). Significant heterogeneity suggests that independent analysis of each species is a more appropriate approach (*Hardy & Thompson, 1998*). A significant moderator further supports analyzing each species separately in a conventional linear *post hoc* model such as a regression. This is in line with the experimental design of separate trials. Linear regressions were thus used to test the effects of mean temperature tested per species on the proportion germinated, proportion established, and *per capita* biomass. Hence, a total of 12 linear regressions were conducted, three for each tested plant species.

## RESULTS

The mean temperature of the warmest quarter and the maximum temperature of the warmest month did not significantly predict the relative frequency of observation for any of the species tested (Fig. 1; Table S1). In the meta-analyses between groups, heterogeneity was significant for all three response variables tested (Table 1). Species and temperature were significant moderators in the meta-analysis for all three response variables (Table 1). Hence, regressions per species were done for each response since each species was tested independently. The germination of *B. rubens*, *P. tanacetifolia*, and *L. platyglossa* was negatively affected by increasing temperatures, while *S. columbariae* was not significantly influenced by temperature (Table 2; Fig. 2). The establishment of *B. rubens* and *P. tanacetifolia* were significantly reduced with increasing temperatures, while

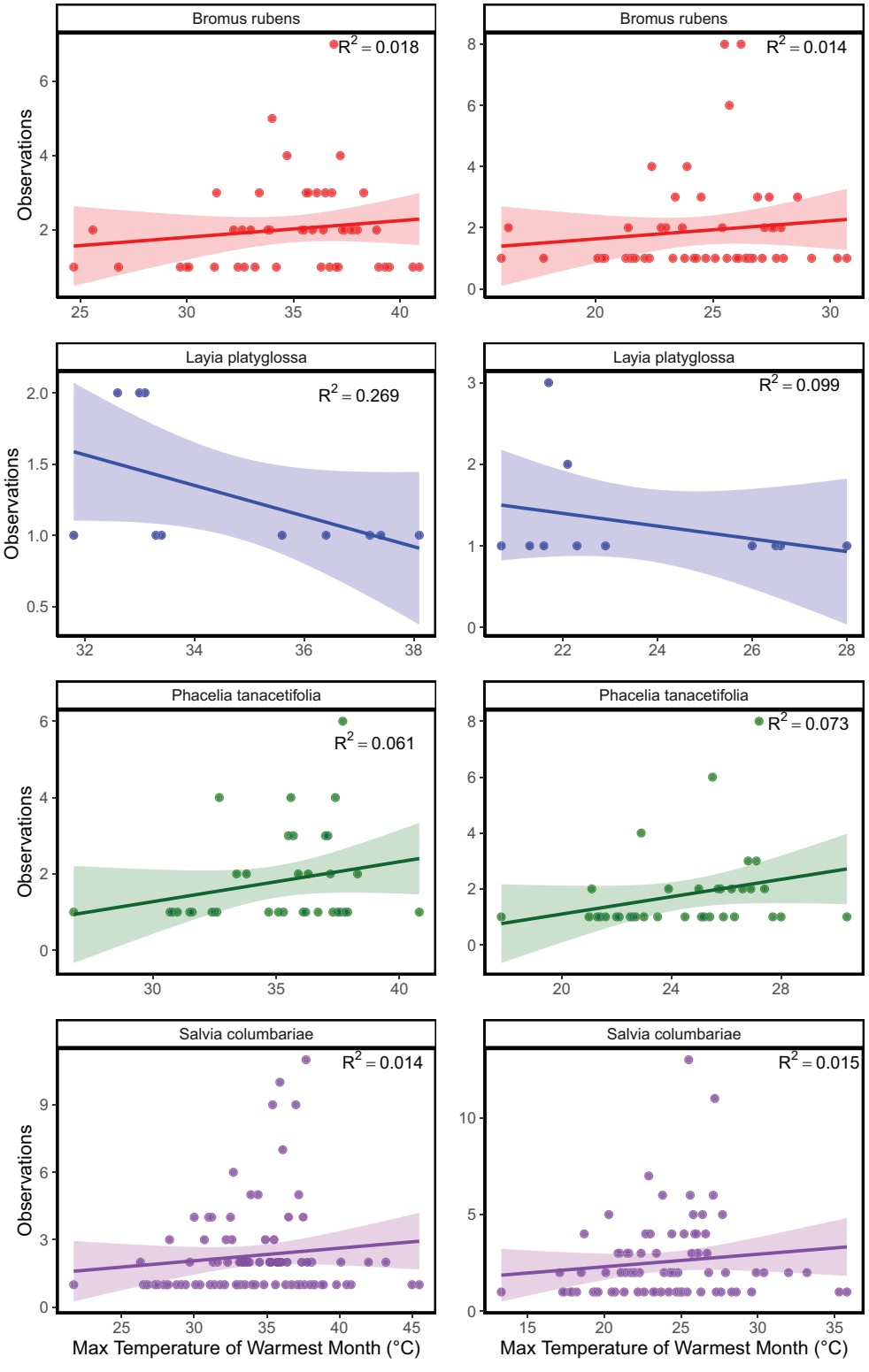

**Figure 1 Linear regression analysis of three native and one invasive annual plant species.** Max temperature from the warmest month and mean temperature from the warmest quarter were acquired from WorldClim and were combined with frequency of species observations from GBIF.

**Table 1 Meta-analyses testing species and temperature as moderators for germination, establishment, and biomass of all tested plant species.**

| Factor | Test | Estimate | df | p-value |
|---|---|---|---|---|
| Germination | Heterogeneity | 211.53 | 31 | **<0.001** |
| | Moderator: Species | 242.63 | 31 | **<0.001** |
| | Moderator: Temperature | 28.42 | 31 | **<0.001** |
| | Moderators: Species*Temperature | 373.06 | 31 | **<0.001** |
| Establishment | Heterogeneity | 138.81 | 31 | **<0.001** |
| | Moderator: Species | 285.11 | 31 | **<0.001** |
| | Moderator: Temperature | 6.17 | 31 | **<0.001** |
| | Moderators: Species*Temperature | 356.21 | 31 | **<0.001** |
| Biomass | Heterogeneity | 94.76 | 31 | **<0.001** |
| | Moderator: Species | 5.91 | 31 | **<0.001** |
| | Moderator: Temperature | 4.39 | 31 | **<0.001** |
| | Moderators: Species*Temperature | 33.32 | 31 | **<0.001** |

Note:
The heterogeneity of each model is reported and the interaction terms for species by temperature. All $p$-values that were significant at $p < 0.05$ are indicated in bold.

*L. platyglossa* and *S. columbariae* were not significantly influenced (Table 2; Fig. 3). Temperature did not significantly influence the *per capita* biomass of the tested plant species (Table 2; Fig. 4).

# DISCUSSION AND CONCLUSION

The effects of temperature on both native and invasive plant species will enhance both current empirical models for community assembly and advance plant-interaction theory in the context of a changing climate. Particularly in dryland ecosystems, temperature can alter the composition of plant communities, creating conditions more favorable for exotic plant invasions. We found support for the hypothesis that temperature directly influences key early-life stage performance measures in common dryland plant species. The effects of increasing temperature were species specific, significantly influencing their germination and establishment. We found that *B. rubens* germination and establishment were significantly reduced at higher temperatures, while only germination was negatively impacted for the native species *L. platyglossa* and *P. tanacetifolia*. We also found that increased temperature niches of these species within Southern California did not influence the total number of reported GBIF observations. Thus, we did not find support for our first prediction that increasing temperatures will negatively influence plant species observations taken from GBIF. We did not find support for our second prediction that increasing temperature negatively influenced all early-life stage performance measures of *B. rubens*. Finally, we found mixed support for our third prediction as higher temperatures negatively impacted germination and establishment of native plant species, while *per capita* biomass was not significantly influenced by increasing temperature. These findings suggest that increasing temperature can influence some key early-life stage measures or specific native and invasive species, providing new insight into the management of these tested species.

**Table 2 Regression analysis of mean experimental temperature on the germination, establishment, and biomass of all tested plant species.**

| Factor | Species | $r^2$ | Moderator | estimate | se | df | t value | p-value |
|---|---|---|---|---|---|---|---|---|
| Germination | Bromus rubens | 0.6064 | Mean temperature | −0.1940 | 0.0659 | 6 | −2.947 | **0.026** |
| | | | Mean temperature$^2$ | 0.0494 | 0.0659 | 6 | 0.750 | 0.482 |
| | Layia platyglossa | 0.7329 | Mean temperature | −0.2118 | 0.0549 | 6 | −3.862 | **0.008** |
| | | | Mean temperature$^2$ | 0.04959 | 0.0549 | 6 | 0.904 | 0.401 |
| | Phacelia tanacetifolia | 0.6569 | Mean temperature | −0.1537 | 0.0469 | 6 | −3.272 | **0.014** |
| | | | Mean temperature$^2$ | 0.0417 | 0.0469 | 6 | 0.887 | 0.409 |
| | Salvia columbariae | 0.5303 | Mean temperature | −0.0103 | 0.0072 | 6 | −1.437 | 0.201 |
| | | | Mean temperature$^2$ | 0.01555 | 0.0072 | 6 | 2.171 | 0.073 |
| Establishment | Bromus rubens | 0.5860 | Mean temperature | −0.1615 | 0.0575 | 6 | −2.812 | **0.031** |
| | | | Mean temperature$^2$ | 0.0449 | 0.0575 | 6 | 0.766 | 0.473 |
| | Layia platyglossa | 0.4868 | Mean temperature | 0.0720 | 0.0331 | 6 | 2.179 | 0.072 |
| | | | Mean temperature$^2$ | −0.0321 | 0.0331 | 6 | −0.971 | 0.369 |
| | Phacelia tanacetifolia | 0.6265 | Mean temperature | −0.1493 | 0.0486 | 6 | −3.075 | **0.022** |
| | | | Mean temperature$^2$ | 0.03791 | 0.0486 | 6 | 0.781 | 0.465 |
| | Salvia columbariae | 0.5403 | Mean temperature | −0.0059 | 0.0044 | 6 | −1.339 | 0.229 |
| | | | Mean temperature$^2$ | 0.0102 | 0.0044 | 6 | 2.294 | 0.062 |
| Biomass | Bromus rubens | 0.3983 | Mean temperature | −0.0163 | 0.0092 | 6 | −1.786 | 0.124 |
| | | | Mean temperature$^2$ | 0.0081 | 0.0092 | 6 | −0.885 | 0.410 |
| | Layia platyglossa | 0.6045 | Mean temperature | −0.0319 | 0.0144 | 6 | −2.217 | 0.068 |
| | | | Mean temperature$^2$ | 0.0296 | 0.0144 | 6 | 2.063 | 0.085 |
| | Phacelia tanacetifolia | 0.4378 | Mean temperature | 0.0127 | 0.0073 | 6 | 1.730 | 0.134 |
| | | | Mean Temperature$^2$ | −0.0095 | 0.0073 | 6 | −1.296 | 0.242 |
| | Salvia columbariae | 0.1021 | Mean temperature | −0.0029 | 0.0216 | 6 | −0.136 | 0.896 |
| | | | Mean temperature$^2$ | 0.0176 | 0.0216 | 6 | 0.815 | 0.446 |

**Note:**
Regressions were run independently for each species and factor to account for heterogeneity between experimental trials. All p-values that were significant at $p < 0.05$ are indicated in bold.

Increasing temperature is one critical component of a changing climate, particularly in dryland ecosystems. In this study, we utilized increasing temperatures to simulate higher abiotic stressors, as many dryland regions within Southern California are experiencing increasingly harsh conditions (*Renwick et al., 2018*; *Scholes, 2020*). An increasing frequency of these higher temperatures can result in more drought events, subsequently resulting in a higher rate of water loss both at an ecosystem and species level (*Reynolds et al., 1999*; *Farooq et al., 2009*). The negative impact of temperature on plant germination suggests thermoinhibition can be an important part of the lifecycle of these species. Thermoinhibition was not directly tested in this study, however it could be a possible explanation as to why higher temperatures resulted in lower germinations of all tested plant species. Thermoinhibition of ungerminated seeds can result in dormancy, reducing the biodiversity of ecosystems (*da Silva et al., 2017*). However, this inhibition can be reversed once more favorable conditions for the target species are reached (*Guo, Shen & Shi, 2020*). In dryland ecosystems, such as the deserts of Southern California, the

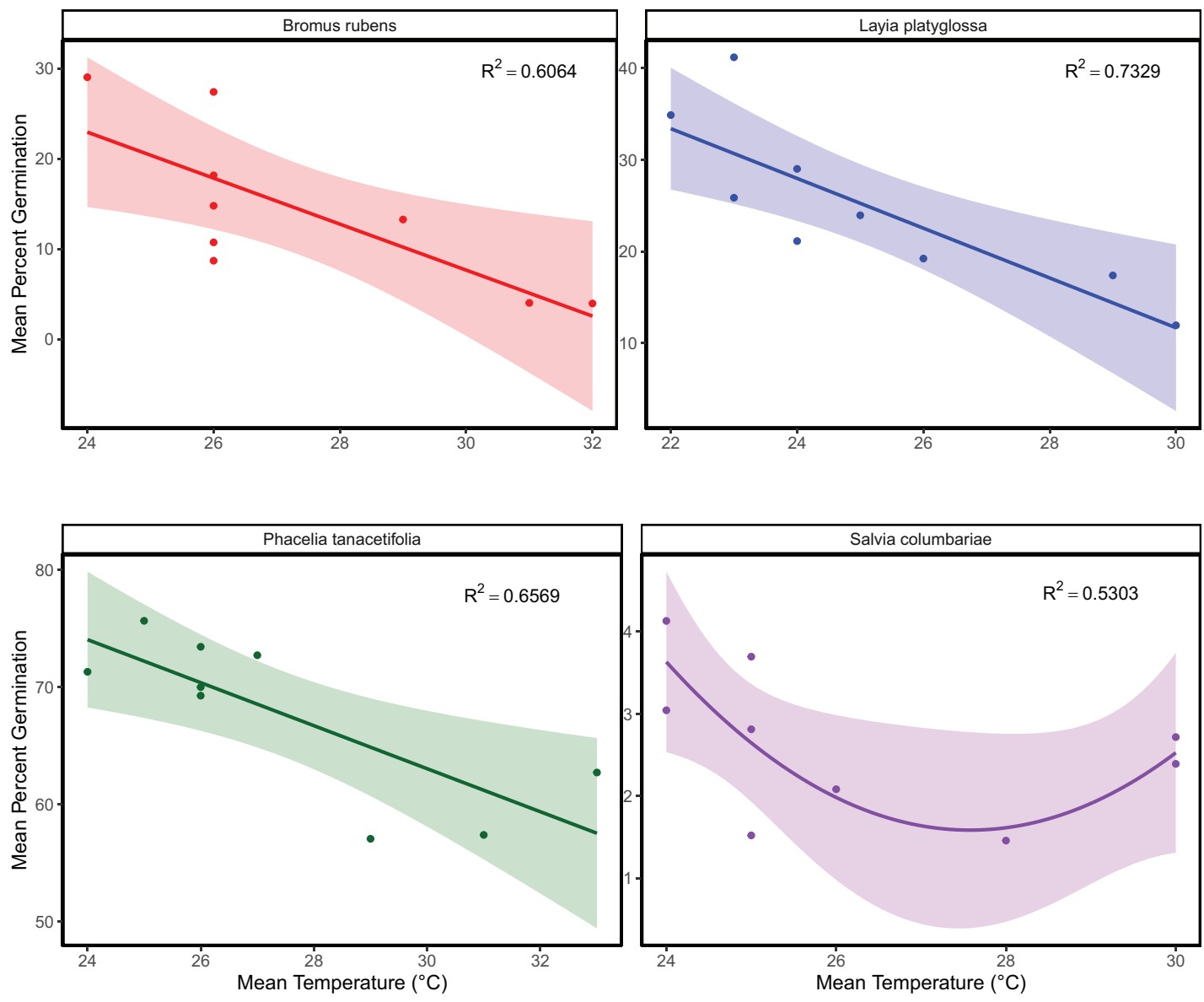

**Figure 2 The relative effects of temperature on the overall germination of native and invasive Southern California plant species.** Germination trials were conducted for each species independently for 6-weeks. Points indicate the mean percent of successfully germinated plant individuals per temperature treatment. Shaded areas show a 95% confidence interval band for the lines of best fit. See Table 1 for complete statistics.

conditions that induce thermoinhibition can become more common as increasing drought events and temperature extremes become more frequent (*Potts et al., 2012*; *Diffenbaugh, Swain & Touma, 2015*), making this an important strategy for the long-term persistence of these species. It is possible that during our study, trials of higher temperature could have reduced the overall availability of water within each pot, thus reducing plant species germination, establishment, and seedling biomass. In addition, light intensity can have a direct impact on the early-life stages of plant development and typically displays a positive correlation with temperature (*Forde, Whitehead & Rowley, 1975*; *Yan et al., 2013*).

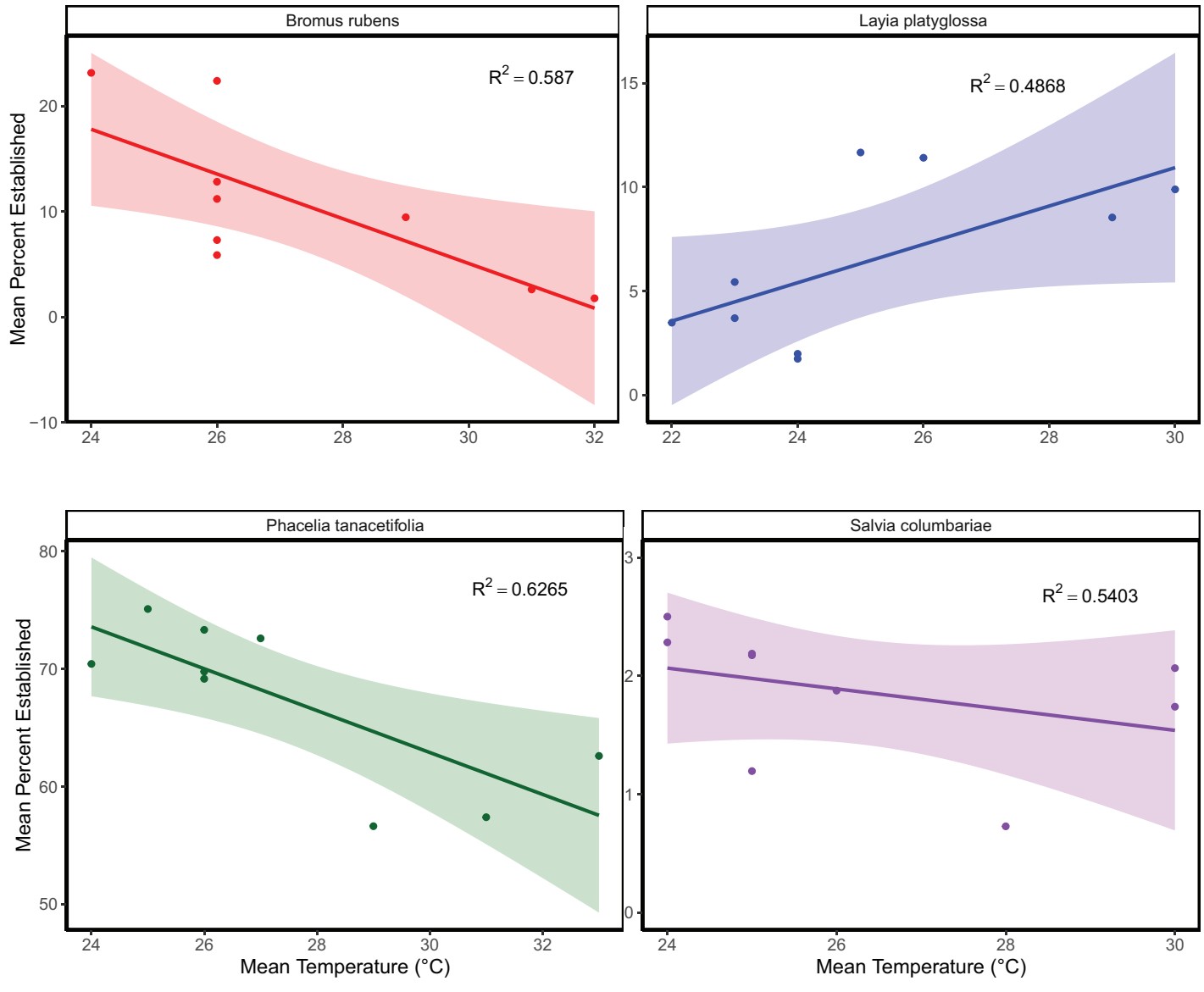

**Figure 3 The relative effects of temperature on the establishment of native and invasive Southern California plant species.** Trials were conducted for each species independently for 6-weeks. Points indicate the mean percent of successfully established individuals for each temperature treatment. Shaded areas show a 95% confidence interval band for the lines of best fit. See Table 1 for complete statistics.

However, for the premise of this experiment, we focused primarily on temperature extremes that would more likely be experienced in drylands as all the tested species can establish populations in these ecosystems. At the current rate of climate change, local plant species may be greatly hindered in their ability to mitigate these increasing temperatures, thus reducing their ability to establish in these ecosystems (*Sosa, Vásquez-Cruz & Villarreal-Quintanilla, 2020*). These increasing temperatures influencing germination and establishment in tandem with one another can impact the overall structure and biodiversity of dryland ecosystems.

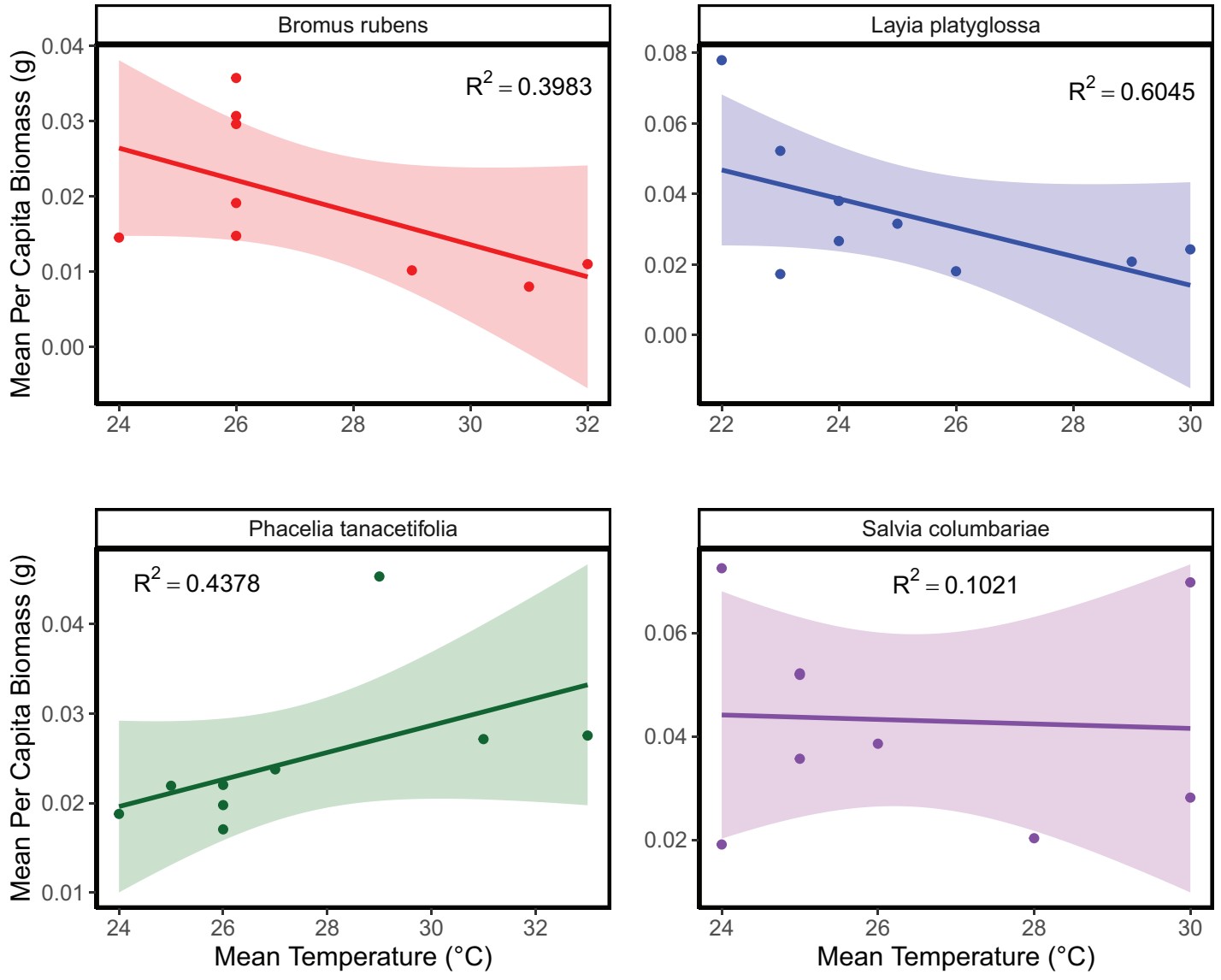

**Figure 4 The relative effects of temperature on the total biomass of native and invasive Southern California plant species.** Trials were conducted for each species independently for 6-weeks. Points indicate the mean *per capita* biomass (g) of established individuals for each temperature treatment. Shaded areas show a 95% confidence interval band for the lines of best fit. See Table 1 for complete statistics.

Understanding key traits of invasive species in all ecosystems is crucial to inform conservation and management. These invasive plant species can outcompete native species as they possess key traits that help them quickly and effectively establish a population, including rapid growth rate, high reproduction, high dispersal rate, and wide tolerance to environmental conditions (*Mathakutha et al., 2019*). Here, we tested only temperature and potential drought conditions as a proxy for a changing climate within dryland ecosystems. The simple climate niche estimates we derived from the WorldClim data suggests that there is a maximum temperature that will impact the ability of *B. rubens* to germinate and establish in an ecosystem. If this invasive species was able to survive at temperatures higher

than its estimated niche, then it may easily outcompete natives in these ecosystems. However, our findings suggest that higher temperatures reduced the germination and establishment of *B. rubens*. Despite the widespread distribution of *B. rubens* across Southern California, this invasive species is not resistant to an increasingly warming climate with our findings suggesting that they will be negatively impacted with increasing temperatures. Since its introduction into North America, this species has rapidly invaded and established populations in several arid ecosystems, negatively affecting the local biodiversity of both plant and animal species through alterations in microhabitat, increased nutrient competition, and altering fire regimens (*Bossard, Randall & Hoshovsky, 2000*; *Hamilton, Holzapfel & Mahall, 1999*; *Brooks, 2000*; *Freeman et al., 2014*). Within these dryland ecosystems, invasive species may be reliant on facilitation from foundational species to successfully establish their populations because of this increased temperature sensitivity. These foundational shrub species facilitate local communities (*Lortie et al., 2020*; *Zuliani et al., 2021*; *Zuliani, Ghazian & Lortie, 2021*), providing benefits to both plant and animal species within Southern California through reducing heat stress by shading and increasing soil moisture (*Prieto, Kikvidze & Pugnaire, 2010*; *Filazzola et al., 2018*, *2020*). However, there is evidence which suggests that as the aridity of these ecosystems further increases, invasive species will no longer be able to benefit from the facilitative effects of shrubs due to global climate change and increased drought events, while native plant species continue to experience these benefits (*Lucero et al., 2022*). In addition to temperature, light intensity and density-dependent interactions can influence the overall establishment of *B. rubens* (*Pik et al., 2020*). Our findings suggest that if increasing temperatures reduce the overall germination and abundance of these invasive plant species more so than native species, then native species may have a competitive advantage over invasive species like *B. rubens* at high temperatures. Several studies have tested the relationship between the native plant species and the invasive *B. rubens* species (*Pik et al., 2020*; *Ghazian et al., 2021*; *Braun et al., 2023*). In studies that tested the impacts of light, seed density, and water level, the native annual species *P. tanacetifolia*, outcompetes *B. rubens*, reducing their germination, establishment, and *per capita* seedling biomass (*Pik et al., 2020*; *Ghazian et al., 2021*; *Braun et al., 2023*). This suggests that the native annual species *P. tanacetifolia* can be utilized for restoration, particularly in areas dominated by *B. rubens*, as this species is more resilient in harsher abiotic conditions (*Braun et al., 2023*). Hence, we suggest that studies focusing on the effects of abiotic conditions on dryland plant communities assess the composition of these species and the relative impacts abiotic factors can have on the germination, establishment, and seedling biomass of invasive and native species. Nonetheless, this study shows that temperature can potentially become a limitation to the early-life stage processes of *B. rubens*, suggesting relatively warmer sites within a region might favor natives. Species-specific responses to increasing temperatures can influence conservation methods as it can determine the timing and location of targeted restoration practices (*Zabin et al., 2022*). By understanding how individual species respond to temperature changes, managers can decide when and where to plant specific species to maximize their establishment and growth success (*Schwartz, 2012*; *Poland et al., 2021*). This can be further supported by the integration of long-term climate data and climate

scenarios, which help predict favorable conditions for native species while minimizing the risk of being outcompeted by invasive species (*Harris et al., 2006*; *Hellmann et al., 2008*).

As one of our tested native plant species demonstrated tolerance to increasing temperatures, selecting specific temperature-resilient native species for mitigation projects may help maintain and restore native plant communities (*Galatowitsch, Frelich & Phillips-Mao, 2009*; *Vitt et al., 2010*). These native species could outcompete invasive plant species in areas with rising temperatures by occupying ecological niches and utilizing resources more effectively, thereby reducing opportunities for invasives to establish and spread (*Čuda et al., 2015*; *Fernández & Hamilton, 2015*). Further emphasis should be placed on temperature as a critical factor for restoration and conservation plans within these dryland ecosystems.

With the increasing frequency of climatic events and invasions by exotic plant species, understanding at least one component of climate change—temperature—can provide valuable information for conservation in dryland systems. Previous studies focusing on early-life performance of dryland native and invasive species have shown similar results. Research conducted has tested the effects of light (*Svriz et al., 2014*; *Pik et al., 2020*; *Nakagawa-Lagisz & Lagisz, 2023*), seed aggregation (*Ghazian et al., 2021*), and water level (*Braun et al., 2023*) on early-life stage performances of both native and invasive plant species. Our findings, as well as the findings from these previous studies, suggest that the native species *P. tanacetifolia* and *B. rubens* suffer the most from increasing temperatures. It is possible that some of the tested native species could outcompete *B. rubens*, as they did not suffer as much as the invasive species did. However, there are other stressors that can be tested, including water and nutrient availability, to further enhance the growing body of literature to support better restoration practices, in the context of highly invasive plant species in dryland ecosystems.

There are several caveats associated with the design of this experiment that could be explored in future temperature experiments with native and invasive annual plant species. Firstly, temperature variation, while significant across tested trials, could be more directly controlled. For instance, alternative methods of heating pots or the use of growth chambers could provide more precise control over daily and seasonal temperature cycles, which are essential for understanding how temperature-driven shifts impact native and invasive species at different life stages (*Beveridge et al., 2024*; *Conneway et al., 2015*). Understanding how a greater and more variable range of temperatures influences plant species can provide managers with insight on how specific abiotic factors impact plant communities, while also influencing when, where, and what specific practices managers could use to promote the restoration of native plant species (*Adler, Dalgleish & Ellner, 2012*). Secondly, the non-random placing of pot could have confounding results on early-life stage measures of all tested plant species. Non-random placing of pots may introduce microclimate variability and edge effect where some pots experience variations in temperature, humidity, and light exposure compared to individuals situated directly under each temperature treatment (*Hartung et al., 2019*; *Ma et al., 2019*). Randomly relocating pots at each treatment within a table weekly can act as a possible means of reducing microclimatic variation and edge effects (*Ma et al., 2019*). Finally, temporal decoupling of species may

have confounded results on the early-life stages of all tested plant species. Testing species at different times could influence photoperiod sensitivity to germination and age of seeds (*Romano & Stevanato, 2020*; *Ettinger et al., 2021*). Conducting all trials for tested native and invasive species simultaneously under each treatment can reduce possible confounding results, simulating more *in-situ* conditions (*Anderson, 2016*).

Further analysis of these species and the relative importance of temperature and other drivers of change, such as timing of events and rainfall, can provide more robust predictive models. We conducted independent 6 week trials to directly assess only early-stage development measures of these plant species. However, many of these species can survive for longer periods of time after germination and can germinate in ecosystems outside drylands. Focusing on the effects of increasing temperatures on established seedlings could provide more insight into their resilience and adaptation to their changing environment. These species typically do not exist independently in nature and experience interspecific interactions. For the purposes of this study, we chose to test temperature independently to remove the effects of these interspecific interactions that could impact the early-life stages of these plant species. Our findings can guide conservation through highlighting future habitats that may be favored more by natives than invasive species. With the decline in dryland ecosystem health and increase in global temperature and aridity, it is essential to study abiotic factors that can impact plant community assembly—particularly with respect to native *vs* invasive plant species.

### Funding

This research was made possible through the Natural Science and Engineering Research Council of Canada (NSERC) grant awarded to Christopher J. Lortie, and the Academic Excellence fund awarded to Mario Zuliani. The funders had no role in study design, data collection and analysis, decision to publish, or preparation of the manuscript.

### Grant Disclosures

The following grant information was disclosed by the authors:
Natural Science and Engineering Research Council of Canada (NSERC).
Academic Excellence.

### Competing Interests

Christopher Lortie is an Academic Editor for PeerJ.

### Author Contributions

- Mario Zuliani conceived and designed the experiments, performed the experiments, analyzed the data, prepared figures and/or tables, authored or reviewed drafts of the article, and approved the final draft.
- Stephanie Haas-Desmarais analyzed the data, prepared figures and/or tables, authored or reviewed drafts of the article, and approved the final draft.

- Laura Brussa performed the experiments, authored or reviewed drafts of the article, and approved the final draft.
- Jessica Cunsolo performed the experiments, authored or reviewed drafts of the article, and approved the final draft.
- Angela Zuliani performed the experiments, authored or reviewed drafts of the article, and approved the final draft.
- Christopher J. Lortie conceived and designed the experiments, analyzed the data, prepared figures and/or tables, authored or reviewed drafts of the article, and approved the final draft.

## Data Availability

All data is available on the Knowledge Network of Biocomplexity (KNB): Mario Zuliani, Laura Brussa, Jessica Cunsolo, Angela Zuliani, & Christopher Lortie. (2021). The effects of varying temperature on the germination of California natives and invasive plant species. Knowledge Network for Biocomplexity. doi: 10.5063/F1VD6WXG.

## Supplemental Information

Supplemental information for this article can be found online at http://dx.doi.org/10.7717/peerj.18794#supplemental-information.

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
