# Peer review of "Controlled temperature contrasts of three native and one highly invasive annual plant species in California"

_PeerJ, doi:10.7717/peerj.18794_

## Round 0.1 · original submission · Major Revisions

Invasive plants represent a critical ecological phenomenon, and understanding how they germinate, grow under various environmental conditions, and compete with native flora is essential. This study offers valuable insights into the invasion dynamics of Bromus rubens. However, certain technical aspects of the article warrant careful attention and refinement. I strongly recommend a thorough review of the reviewers' suggestions, along with a thoughtful consideration of each recommendation. If you find reason to disagree with particular suggestions, providing a clear and well-reasoned justification for your perspective would be advantageous.

Reviewer 1 ·

Basic reporting

Writing could be improved for clarity. Here are a few examples.

31: predicting invasive species?
35-37: "These species..." Delete this sentence
39: "microclimatic effects" is unclear. I suggest changing to "effects of temperature variation..."
45: "will likely"
47: "our native annual plant species"
48: "can favor at least one native species..."
53: Please ensure that topic sentences are used that lead the reader to know what to expect in the paragraph.
55: I don't understand the opening sentence.
57: like what?
62: which balance?
131: benefit? This contradicts prediction 1. Also, what evidence exists that this invasive species has a broader thermal niche than your native species? Which native species are you using? This prediction is not sufficiently justified in the intro.
134: How is this different from #1?
175: Separate into 2 paragraphs, one for species, one for seed collection.
178: Delete this opening clause; it is undermined by your field collection of B. rubens.
189: what is "successful"?
195: what does this last clause mean?
197: overlapped? How? I don't understand this sentence.
230: Define acronym and justify why this is a good approach.
243: I don't understand this.

Experimental design

131: benefit? This contradicts prediction 1. Also, what evidence exists that this invasive species has a broader thermal niche than your native species? Which native species are you using? This prediction is not sufficiently justified in the intro.
134: How is this different from #1?
181: Does the airtight environment induce anoxia and potentially interfere with seed/seedling performance?
198: I'm not sure I am convinced of this.
199: How were pots situated spatially in the greenhouse?? Seems non-random to me.
205: I'm not sure that MiracleGro and sand approximate desert soils, which are famously poor in nutrients and OM.
207: This is a high degree of variation! 50% variation among pots could count as a drought treatment!! How was this accounted for in the study?? How exactly was water delivered? Why so much variability??
207: Temperature and drought covaried in this study. Thus, I am concerned that plant responses may have been due to drought, not temperature.
210: This min and max don't tell us much about growing conditions. What are the average temps of each lamp?? Also, state the ambient temperature in the greenhouse, as plants experienced this environment 50% of the time.
230: Define acronym and justify why this is a good approach.
246: I don't see how this approach validates the 22-33 C range used in the greenhouse?
254: I don't understand why meta-analysis was used instead of linear mixed effects models.
271: I do not understand the incorporation of meta-analysis.
280: The discussion must mention the following caveats: 1)Temperature variation in greenhouse was quite modest; 2)Temperature and drought were auto-correlated in the greenhouse so that plant responses to temperature may be confounded by concomitant drought.
3) Taxonomic replication of native and invasive species is very coarse; 4) Non-random placement of plants in the greenhouse study may have confounded results; 5) Temporal decoupling caused by the serial nature of experiential design in the greenhouse (one trial followed by another instead of all occurring simultaneously) may have confounded results; 6) watering varied by 50% in the greenhouse!! Watering variability may have confounded results, especially because plants were not randomly arranged in greenhouse benches.
291: I don't think I agree; this conclusion does not follow the result described in the previous sentence.
295: I don't really understand the study predictions and why they were made.
299: I'm not sure this last sentence is true, as plant responses are species-specific and no effort is made to connect plant responses to the community level. Also, the study is essentially univariate, and temperature is not the only thing that matters in plant communities.
301: I don't think the two temperature treatments count as "extremes" for southern California. Actually, the temperature variation here (22-33 C) is quite modest relative to the range of field conditions potentially experienced by the study species (Fig. 2 supports this)! Perhaps it's actually quite impressive that plants showed any responses at all to temperature? Perhaps this study suggests that the study species can be very sensitive even to modest changes in temperature?
301: 22-33 C is not an "intense climatic event" for California drylands. For context, the average minimum and maximum temps recorded annually in Fresno (near Carrizo) are 4.4 and 37 C, respectively. Thus, temp variability explored here in the greenhouse didn't even flirt with the low end, and barely flirted with the high end of these averages -- and this is in Fresno, whose climate is quite tame compared to other California drylands (e.g., the Mojave area). I’m not convinced that these findings are particularly relevant to climate change.
306: Again, I'm not sure that I trust the results of the greenhouse study presented here, due to unaccounted for potential for autocorrelation of temperature stress and drought stress, non-random placement of plants in the greenhouse, temporal decoupling caused by the serial nature of experiential design in the greenhouse (one trial followed by another instead of all occurring simultaneously), watering varied by 50% in the greenhouse -- could be construed as a drought treatment!!
333: the study didn't test drought but was certainly confounded by drought. One strategy might be to get away from claiming to test of temperature and instead own that the study actually performed a test of coupled temp-drought?? This might make sense, as these variables are also autocorrelated in the field. This approach could allay concerns about experimental design.
341: Not sure the data support this.
385: Yes, but Bromus could just as easily be more tolerant of a wider variety of environmental stressors than natives?? Temperature is not the only thing that matters.
393: confounded by a few other factors
Fig. 3: The main manuscript claims that all tested species suffered from increasing temperature. Salvia doesn't completely fit this description; see non-linear relationship.
Fig 4: I see different effects of temp on germination and establishment. Establishment is likely a more sensitive life history transition than germination for lambda. Might be a good discussion point.
Fig. 5: I suggest reporting test statistics for all regressions in this and other figures so we can easily see which relationships are and aren't significant. For non-significant regressions, maybe remove confidence band and show with a dashed line??
Table 1: The use of meta-analysis is not sufficiently justified, I think.

Validity of the findings

Findings often overstated , conclusions often not supported by data.

Additional comments

Manuscript #107472, “Controlled temperature contrasts of California native
species and a highly invasive plant species” combined an experimental greenhouse approach and an observational thermal niche modelling approach to compare and contrast the responses of three native and one invasive plant species to thermal stress. The authors conclude that plant responses to temperature stress are species-specific.

The greenhouse study has some major (potentially fatal) methodological issues, the results are often strongly overstated, conclusions are often not supported by the data, and the writing could use revision for clarity. Some of these issues can be addressed in a revised manuscript, but the methodological flaws are likely unfixable and so must be very clearly acknowledged in the discussion so that readers can more easily evaluate the merit of this work for themselves.

31: predicting invasive species?
35-37: "These species..." Delete this sentence
39: "microclimatic effects" is unclear. I suggest changing to "effects of temperature variation..."
45: "will likely"
47: "our native annual plant species"
48: "can favor at least one native species..."
49: I'm not sure what this study teaches us about thermal ecology because the variability used here was so modest, and because temperature was confounded with a number of other factors in the greenhouse.
53: Please ensure that topic sentences are used that lead the reader to know what to expect in the paragraph.
55: I don't understand the opening sentence.
57: like what?
62: which balance?
131: benefit? This contradicts prediction 1. Also, what evidence exists that this invasive species has a broader thermal niche than your native species? Which native species are you using? This prediction is not sufficiently justified in the intro.
134: How is this different from #1?
175: Separate into 2 paragraphs, one for species, one for seed collection.
178: Delete this opening clause; it is undermined by your field collection of B. rubens.
181: Does the airtight environment induce anoxia and potentially interfere with seed/seedling performance?
189: what is "successful"?
195: what does this last clause mean?
197: overlapped? How? I don't understand this sentence.
198: I'm not sure I am convinced of this.
199: How were pots situated spatially in the greenhouse?? Seems non-random to me.
205: I'm not sure that MiracleGro and sand approximate desert soils, which are famously poor in nutrients and OM.
207: This is a high degree of variation! 50% variation among pots could count as a drought treatment!! How was this accounted for in the study?? How exactly was water delivered? Why so much variability??
207: Temperature and drought covaried in this study. Thus, I am concerned that plant responses may have been due to drought, not temperature.
210: This min and max don't tell us much about growing conditions. What are the average temps of each lamp?? Also, state the ambient temperature in the greenhouse, as plants experienced this environment 50% of the time.
230: Define acronym and justify why this is a good approach.
243: I don't understand this.
246: I don't see how this approach validates the 22-33 C range used in the greenhouse?
254: I don't understand why meta-analysis was used instead of linear mixed effects models.
271: I do not understand the incorporation of meta-analysis.
280: The discussion must mention the following caveats: 1)Temperature variation in greenhouse was quite modest; 2)Temperature and drought were auto-correlated in the greenhouse so that plant responses to temperature may be confounded by concomitant drought.
3) Taxonomic replication of native and invasive species is very coarse; 4) Non-random placement of plants in the greenhouse study may have confounded results; 5) Temporal decoupling caused by the serial nature of experiential design in the greenhouse (one trial followed by another instead of all occurring simultaneously) may have confounded results; 6) watering varied by 50% in the greenhouse!! Watering variability may have confounded results, especially because plants were not randomly arranged in greenhouse benches.
291: I don't think I agree; this conclusion does not follow the result described in the previous sentence.
295: I don't really understand the study predictions and why they were made.
299: I'm not sure this last sentence is true, as plant responses are species-specific and no effort is made to connect plant responses to the community level. Also, the study is essentially univariate, and temperature is not the only thing that matters in plant communities.
301: I don't think the two temperature treatments count as "extremes" for southern California. Actually, the temperature variation here (22-33 C) is quite modest relative to the range of field conditions potentially experienced by the study species (Fig. 2 supports this)! Perhaps it's actually quite impressive that plants showed any responses at all to temperature? Perhaps this study suggests that the study species can be very sensitive even to modest changes in temperature?
301: 22-33 C is not an "intense climatic event" for California drylands. For context, the average minimum and maximum temps recorded annually in Fresno (near Carrizo) are 4.4 and 37 C, respectively. Thus, temp variability explored here in the greenhouse didn't even flirt with the low end, and barely flirted with the high end of these averages -- and this is in Fresno, whose climate is quite tame compared to other California drylands (e.g., the Mojave area). I’m not convinced that these findings are particularly relevant to climate change.
306: Again, I'm not sure that I trust the results of the greenhouse study presented here, due to unaccounted for potential for autocorrelation of temperature stress and drought stress, non-random placement of plants in the greenhouse, temporal decoupling caused by the serial nature of experiential design in the greenhouse (one trial followed by another instead of all occurring simultaneously), watering varied by 50% in the greenhouse -- could be construed as a drought treatment!!
329: all?
333: the study didn't test drought but was certainly confounded by drought. One strategy might be to get away from claiming to test of temperature and instead own that the study actually performed a test of coupled temp-drought?? This might make sense, as these variables are also autocorrelated in the field. This approach could allay concerns about experimental design.
341: Not sure the data support this.
385: Yes, but Bromus could just as easily be more tolerant of a wider variety of environmental stressors than natives?? Temperature is not the only thing that matters.
393: confounded by a few other factors
Fig. 3: The main manuscript claims that all tested species suffered from increasing temperature. Salvia doesn't completely fit this description; see non-linear relationship.
Fig 4: I see different effects of temp on germination and establishment. Establishment is likely a more sensitive life history transition than germination for lambda. Might be a good discussion point.
Fig. 5: I suggest reporting test statistics for all regressions in this and other figures so we can easily see which relationships are and aren't significant. For non-significant regressions, maybe remove confidence band and show with a dashed line??
Table 1: The use of meta-analysis is not sufficiently justified, I think.

·

Basic reporting

Find the minor correction below;
Line 1: Controlled temperature contrasts of selected native and a highly invasive annual plant
species in California
Line 41-42: Check with your results, it different from your results
Line 64: mention the invaded species.
Line 65: mention the extreme weather conditions.
Line 68: mention how strongly influence whether negative or positive.
Line 89-90: Mention how it can influence plant association and establishment.
Line 94: check the year of the citation
Line 95: mention a few examples of stress amelioration
Line 96-97: give a reference
Line 106: Start in the new paragraph and place this sentence as a second sentence.
Line 110: Bradley et al. 2013 is not in the reference list
Line 112: Abatzglou & Kolden 2011, is not in the reference list
Line 159: Introduce the full scientific name first. Bromus rubens L.; Layia platyglossa (Fisch. & C.A.Mey.) A.Gray; Phacelia tanacetifolia Benth.; Salvia columbariae Benth.;
Line 164: The year of the reference is not correct with the reference list in Marty & BassiriRad 2013
Line 183: Did they do any dormancy break treatment after storing 8℃?
Line 203: the depth of the seed sown seems too high.
Line 207: In the introduction cited that the annual precipitation in a particular area is 25 mm, Is there any concern about applying 50- 75 mL of water?
Line 215: did you calculate the seed germination in the Petri dish or any germination test method to validate each species' germination?
Line 238: the citation is not compatible with the reference list (Pearson et al. 2001).
Line 246: mention the temperature and humidity levels used in your experiment
Line 297-299: idea of the sentence is not clear

Major comments
Why did you select southern California instead of Canada, because Bromus rubense is also present in Canada?
Discuss the native and invasive plant competition in one para in the introduction (eg. Allelopathy etc…)
In Figure 3, How did mean germination in decimal numbers? It's better to compare it with the control experiment.
Discuss why doe P. tanacetifolia has a positive correlation with biomass.
Suggest to include figure 01 in appendixes
Some graphs show up to 30 ℃ and some are 32℃. Axis value should be equal and same scale in both axes because it is easy to understand their behaviour
However, according to the Salvia columbariae in Figure 01, it presents more than 32℃. Thus, suggest showing the X axis more than 32℃.
Write the conclusion separately.
It’s better to show the Table 02 values on the figures and Move Table 02 to the appendix.

Experimental design

Germination should be compared with the control. I think the control experiment of gemination is missing here. The germination test should proceed.

Validity of the findings

The conclusion should be written separately. Literatures are clearly stated. Check the ethical clearance of native species using any permission.

Reviewer 3 ·

Basic reporting

The manuscript is written in clear and professional English, which makes it easy to follow. The language used is precise and unambiguous, which helps to effectively convey the complexity of the topic. The authors provided an adequate overview of the existing literature in the field of arid ecosystems and invasive species research. The inclusion of relevant studies, such as those dealing with the effects of temperature, the behavior of invasive species, and the dynamics of dryland ecosystems, supports the hypotheses presented in the paper.
The article follows a standard structure, including sections such as Introduction, Methods, Results and Discussion. However, a more detailed presentation of the results is suggested to ensure a clearer understanding of the data. Figures and tables are used appropriately to present experimental findings. The study is well connected to the hypotheses set out in the introduction, and the presented results are relevant to answering those hypotheses. Regarding the set hypotheses, I ask the authors to clarify do the first and third predictions essentially state the same idea?

Experimental design

The research is within the aims and scope of the journal. The research question is well defined, focusing on how temperature influences early-life stage performance in both native and invasive plant species in dryland ecosystems. This is a relevant topic, particularly in the context of climate change and its impact on species competition and ecosystem dynamics. The investigation was conducted in controlled greenhouse environment which is an appropriate method to test the effects of temperature on plant performance. Independent trials for each species were conducted with adequate replication. The study conforms to ethical standards, as it involves plant species with no mention of ethical concerns related to endangered or protected taxa. The methods are described in sufficient detail to allow for replication by other researchers. Some objections and suggestions have been inserted directly into the text document itself.

Validity of the findings

The research presents meaningful data on temperature impacts on native and invasive species, which contributes to broader understanding of plant community dynamics. The conclusions are given at the end of the Discussion and are well-stated and directly linked to the original research question.

Annotated reviews are not available for download in order to protect the identity of reviewers who chose to remain anonymous.

---

## Round 0.2 · Minor Revisions

Your article is close to being accepted by PeerJ, but it requires minor improvements before publication. Please carefully read the reviewer's comments and consider each of them. If you disagree with any specific suggestion, please provide clear and well-reasoned justifications to support your perspective.

Reviewer 1 ·

Basic reporting

The author's revisions for clarity have greatly improved the manuscript and cleared up my most salient concerns, most of which arose because of unclear or inaccurate phrasing. The revised version addresses all comments carefully and respectfully, and I have no further comments at this time.

Experimental design

The author's revisions for clarity have greatly improved the manuscript and cleared up my most salient concerns, most of which arose because of unclear or inaccurate phrasing. The revised version addresses all comments carefully and respectfully, and I have no further comments at this time.

Validity of the findings

The author's revisions for clarity have greatly improved the manuscript and cleared up my most salient concerns, most of which arose because of unclear or inaccurate phrasing. The revised version addresses all comments carefully and respectfully, and I have no further comments at this time.

·

Basic reporting

The authors have already addressed the acceptable comments in the revision.

Experimental design

It suggested adding the sub-topic on Research limitations. And add the answers to my comments on the control experiment and water application. Because the rainfall and the temperature of collected seeds differ from the experimental station as they add 75 ml and cold weather.

"Line 207: In the introduction cited that the annual precipitation in a particular area is 25 mm, Is there any concern about applying 50- 75 mL of water?

 Thank you. We have now specified in the manuscript that approximately 75mL of water was used per pot during these trials (L228-229).

Line 215: did you calculate the seed germination in the Petri dish or any germination test method to validate each species' germination?

 Thank you for the comment. We did not conduct germination validation for these seeds. We have however provided average seed viabilities per species from the literature in the Methods (L177-178; 188-195).
"

Validity of the findings

Still, they need to correct the y-axis in Figures 2 and 3. And they mentioned that they corrected it. However, I could not see it in the last version of the article attached.

"In Figure 3, How did mean germination in decimal numbers? It's better to compare it with the control experiment.

 Thank you. The y-axis of this figure is mean percent germinated. So a decimal of 0.3 indicates that 30% of seeds germinated. We have now made this clearer in the figure legend.
"

Reviewer 3 ·

Basic reporting

I have reviewed the manuscript and do not have any additional comments or suggestions for improvement.

Experimental design

I have reviewed the manuscript and do not have any additional comments or suggestions for improvement.

Validity of the findings

I have reviewed the manuscript and do not have any additional comments or suggestions for improvement.

---

## Round 0.3 · accepted · Accept

I appreciate your constructive attitude toward the reviewers' suggestions and improving your article based on their suggestions. I believe your manuscript is now ready for publication. But, you should check two references and correct them (In line 401 Galatowitsch et al. 209; In line 73 Vleeshouwser et al. 1995) before publication. We look forward to your next article.

·

Basic reporting

The authors addressed the raised points.

Experimental design

Nothing new to report

Validity of the findings

Nothing new to report

Additional comments

References should correct,

Line 401 - Galatowitsch et al. 209 ???
Line 939 vs line 73 - Check the spellings of the citation

Suggest to recheck the rest of the references and citations